# Comparative Performances of Beneficial Microorganisms on the Induction of Durum Wheat Tolerance to Fusarium Head Blight

**DOI:** 10.3390/microorganisms9122410

**Published:** 2021-11-23

**Authors:** Zayneb Kthiri, Maissa Ben Jabeur, Kalthoum Harbaoui, Chahine Karmous, Zoubeir Chamekh, Fadia Chairi, Maria Dolores Serret, Jose Luis Araus, Walid Hamada

**Affiliations:** 1Laboratory of Genetics and Cereals Breeding, National Institute of Agronomy of Tunisia, Carthage University, Tunis 1082, Tunisia; maissa.benjabeur@hotmail.com (M.B.J.); karmouschahine@gmail.com (C.K.); w_hamada@yahoo.com (W.H.); 2High School of Agriculture of Mateur, Department of Plant Sciences, Carthage University, Mateur 7030, Tunisia; harbaoui_kalthoum@yahoo.fr; 3National Institute of Agricultural Research of Tunisia, Field Crop, Carthage University, Tunis 2049, Tunisia; zoubeirchamek@gmail.com; 4Integrative Crop Ecophysiology Group, Plant Physiology Section, Faculty of Biology, University of Barcelona, 08028 Barcelona, Spain; fadia.chairi@hotmail.fr (F.C.); dserret@ub.edu (M.D.S.); jaraus@ub.edu (J.L.A.); 5AGROTECNIO (Center of Research in Agrotechnology), University of Lleida, 25198 Lleida, Spain

**Keywords:** durum wheat, grain yield, Fusarium head blight, bioagents, isotope

## Abstract

Durum wheat production is seriously threatened by Fusarium head blight (FHB) attacks in Tunisia, and the seed coating by bio-agents is a great alternative for chemical disease control. This study focuses on evaluating, under field conditions, the effect of seed coating with *Trichoderma harzianum*, *Meyerozyma guilliermondii* and their combination on (i) FHB severity, durum wheat grain yield and TKW in three crop seasons, and (ii) on physiological parameters and the carbon and nitrogen content and isotope composition in leaves and grains of durum wheat. The results indicated that the treatments were effective in reducing FHB severity by 30 to 70% and increasing grain yield with an increased rate ranging from 25 to 68%, compared to the inoculated control. The impact of treatments on grain yield improvement was associated with higher NDVI and chlorophyll content and lower canopy temperature. Furthermore, the treatments mitigated the FHB adverse effects on N and C metabolism by resulting in a higher δ^13^Cgrain (^13^C/^12^Cgrain) and δ^15^Ngrain (^15^N/^14^Ngrain). Overall, the combination outperformed the other seed treatments by producing the highest grain yield and TKW. The high potency of seed coating with the combination suggests that the two microorganisms have synergetic or complementary impacts on wheat.

## 1. Introduction

Wheat is among the major cereal crops produced in the world. The FAO’s latest information points to a global wheat production of 776.7 million tons for 2021–2022, which forms one-third of the world’s total cereal production [1,2].

Wheat diseases are widely causing economic losses, which make them a major limiting factor to wheat production worldwide. Fusarium head blight (FHB), which can be caused by many kinds of Fusarium species, including *F. graminearum*, *F. culmorum*, *F. poae*, *F. avenaceum*, *F. sporotrichioides* and *Microdochium nivale*, is a chronic disease in wheat production in semi-arid regions worldwide that reduces quality and crop yield [3,4]. For example, durum wheat yield losses were reported to reach 52% in Australia, 50% in USA, 46% in Iran and 44% in Tunisia [5]. Infection of seeds may decrease seed germination, reduce emergence and cause a post-emergence blight of seedlings, thus, leading to a less dense plant stand [6].

The control of these fungal diseases is mainly through a large-scale application of chemical fungicides [7]. However, this method is limited by the emergence of resistant pathogens, the adverse effects on the environment and human health [6] and the expensive costs related to the broad foliar application of these fungicides.

The seed coating technique requires few applied agents and is, thus, an alternative to reduce the pesticides applied to crops over a large area, which saves time and expenses for farmers while avoiding the uncertainty of effectiveness of such a broad distribution [8]. In addition, the use of chemical fungicides as seed coatings could not effectively control soil-borne pathogens since they do not last much beyond the seedling stage due to their rapid natural degradation [9]. An alternative approach is the application of seed coating with beneficial microorganisms, which might be able to (i) counteract the pathogenic and mycotoxigenic potential of natural populations of Fusarium surrounding the seed in infected soils, (ii) induce systemic plant resistance against pathogens [10] and (ii) maintain the symbiotic relationship between beneficial microorganisms and the plant along the growth cycle. In fact, seed biopriming using beneficial microorganisms as seed coating agents was used to induce germination, improve the emergence of seedlings and induce mechanisms of plant disease resistance [11,12]

*Meyerozyma guilliermondii* is a yeast reported as a biocontrol agent and inducer of plant resistance [13,14]. *Trichoderma harzianum* is a beneficial fungus known for its biocontrol capacity against soil-borne pathogens and its ability to trigger plant growth [15,16].

In our previous study, we reported that seed coating with *M. guilliermondii* and *T. harzianum* S. INAT stimulate seed germination, induce plant resistance against *F. culmorum* and improve plant growth and photosynthesis in durum wheat under controlled conditions [11,16,17]. Despite their potential under controlled conditions, their biocontrol efficiency against FHB disease of durum wheat under field conditions could be less impressive. Therefore, we explored the effect of seed coating with *T. harzianum*, *M. guilliermondii* and their combination, compared to the commercial product Panoramix, on (a) Fusarium head blight severity and yield components under field conditions and (b) carbon and nitrogen metabolism.

## 2. Materials and Methods

### 2.1. Plant Materiel

A Tunisian variety of durum wheat (*Triticum turgidum* L. subsp. Durum), ‘Karim’ was used in this study; the seeds were chosen as free from chemicals and were stored at room temperature (20–24 °C). This variety is known for its sensitivity to *Fusarium* [18] and is one of the most used durum wheat cultivars in Tunisia [19]).

### 2.2. Seed Coating Treatment

Before sowing, the seeds were coated with *M. guilliermondii* INAT (KU710283), *T. harzianum* (KU710282), and the combination *M. guilliermondii-T. harzianum* as described previously [11,20].

The coating product Agicote Rouge T17 (AEGILOPS Applications, France) was used, containing propane-1,2-diol (5–10%), polyethylene glycol mono (tristyrylphenyl) ether (5–10%) and 1,2-benzisothiasol 3(2H)-one (0.0357). The coating technique for each treatment consisted of mixing 40 μL of the coating product Agicote Rouge T17 (AEGILOPS Applications, France) with 400 μL of either the suspension of *T. harzianum* culture at a concentration of 10^6^ spores·mL^−1^ or the suspension of *M. guilliermondii* culture at a concentration of 10^6^ spores·mL^−1^ (water was used as a control). The combination *M. guilliermondii-T. harzianum* was realized by adding 200 µL of each microorganism suspension at the same concentration of 10^6^ spores·mL^−1^. Then, the coating mixture was applied progressively to 10 g of wheat seeds in a continuous rotation until complete adhesion and absorption to assure homogeneous distribution of the coating mixture among seeds. The effect of the coating product was evaluated in the laboratory prior to the evaluation of the effect of coating seeds in fields. The inertness of the coating product was assured on seed germination and seedling growth.

Additionally, the product “Panoramix” was used as reference. “Panoramix” is a biological seed dressing, marketed by “Koppert”, consisting of a combination of microorganisms and additives which promote plant growth. This product is composed of Mycorrhiza (>10 propagules/mL), *Trichoderma* spp. (>1 × 10^7^ CFU/mL) and Bacillus spp. (2 × 10^7^ CFU/mL) which colonizes the roots and protects the crop during the entire cultivation season [21].

### 2.3. Experimental Design for Field Trials

The experiments were conducted at the Regional Field Crops Research Center (CRGC) experimental station in Oued-Beja (36.73° N, 9.23° E), located in the sub-humid bioclimatic zone of Tunisia, for three years; 2017–2018 and 2019 under rainfed conditions (Table 1). The soil type of the experimental area is mostly clay loam with pH 7.2 (Table 2). A complete random block design with three replicated plots was used. The elementary plot size was 1 × 3 m spaced by 1.5 m. Each plot consisted of 5 rows, with a row spacing of 0.15 m.

The sowing was carried out in the first week of December with a sowing density of 250 seeds/m^2^. Nitrogen (ammonium nitrate) was applied at 25 kg N/ha at sowing and at the stem elongation stage.

### 2.4. Inoculum Production and Inoculation

*F. culmorum* inoculum was prepared by producing *F. culmorum* (Wm.G.Sm.) Sacc. macroconidia. For this purpose, barley grains were soaked in water overnight, excess water was drained off, and grains were autoclaved. PDA plates colonized with *F. culmorum* were added to the autoclaved barley grains and incubated for 3 weeks at 25 °C.

The conidial suspension of *F. culmorum* was prepared by air-drying the colonized barley grains on filter paper, then grounding them in a laboratory mill and filtering through a 2-mm sieve. A final concentration of the conidial suspension of *F. culmorum* inoculum was set at 1 × 10^6^ conidial mL^−1^.

Inoculation of wheat spikes was carried out when the plants were at the mid-flowering growth stage GS 65 (Figure 1) [22]; wheat spikes were inoculated with 1 × 10^6^ conidia mL^−1^ of *F. culmorum* conidial suspension by spray using a CO_2_-pressurized knapsack sprayer, while control plants were sprayed with distilled water.

### 2.5. Disease Assessment

At 20 days post-inoculation, which corresponds to the early milk stage (GS 73), the disease symptoms were visually assessed in every plot (Figure 1). In each plot, 100 randomly selected heads were counted for symptomatic spikelets. The disease severity was calculated as the percentage of symptomatic spikelets per average number of spikelets [23].

### 2.6. Effect of Seed Coating on Yield Components and Physiological Parameters

At the early milk stage (GS 73), five leaves within each plot were selected for non-destructive measurements of leaf chlorophyll content, using a portable meter (SPAD 502 plus, Minolta, UK), and stomatal conductance of the flag leaf with a leaf porometer (Decagon, Pullman, WA, USA). In addition, the following was performed for each plot at the canopy level, canopy normalized difference vegetation index (NDVI) using a spectroradiometer (GreenSeeker@Trimble, Westminster, CO, USA) and canopy temperature using an infrared thermometer (Fluke, Everett, WA, USA). At harvest, one m^2^ in the central part of each plot was hand-harvested. Then, grains were collected using a shredder (Wintersteiger, LD-180, Ried im Innkreis, Austria), and grain yield (GY, Mg ha^−1^) was measured. The thousand kernel weight (TKW, g) was determined for one thousand grains counted by a seed counter Numigral X5 (CHOPIN Technologies, Villeneuve La Garenne, France).

### 2.7. Effect of Seed Coating on Total Nitrogen and Carbon Content and Stable Carbon and Nitrogen Isotope Composition

The total N and C content and stable nitrogen isotope signature in the dry matter of the mature grains and flag leaf sampled from each plot of the third field trial (2018) at the late milk stage (GS 77) were analyzed at the laboratory (University of Barcelona). Approximately 1 mg of each sample and reference materials were weighed into tin capsules and measured with an elemental analyzer (Flash1112EA; Thermo Finnigan, Bremen, Germany) coupled with an isotope ratio mass spectrometer (Delta CIRMS, Thermo Finnigan, Bremen, Germany) operating in continuous flow mode to determine the total C and N content and the stable carbon (^13^C/^12^C) and nitrogen (^15^N/^14^N) isotope ratios. The ratios were expressed in δ notation, as δ^13^C = (^13^C/^12^C) sample/(^13^C/^12^C) standard ^−1^, where sample refers to the plant material and standard to Pee Dee Belemnite (PDB) calcium carbonate, and as δ^15^N = (^15^N/^14^N) sample/(^15^N/^14^N) standard ^−1^, where sample refers to plant material and standard refers to N_2_ in air.

### 2.8. Statistical Analyses

The effects of the treatments and years and their interaction on FHB severity and yield components were determined through a two-factor (treatment × year) analysis of variance (ANOVA) with RStudio 1.1.463 (R Foundation for Statistical Computing, Vienna, Austria). The effects of the treatments on physiological traits, yield components and grain stable isotopic compositions were determined through a one-factor ANOVA (treatment). The least significant difference (LSD) test was used to assess the differences between the treatment means.

## 3. Results

### 3.1. Climatic Features and Sources of Variances of Three Years of Study

The severity of Fusarium head blight differed considerably among years, reflecting climatic effects. The data in Table 1 show that the experimental season 2019 is the most favorable FHB compared to the others. It was characterized by a higher number of annual precipitation and lower maximal temperatures. By contrast, the experimental season 2017 was characterized by a lower amount of precipitation and a higher maximal temperature. Another difference between the two seasons is that wheat plants received 106.2 mm of annual precipitations in May of 2019, while zero precipitation was recorded for the same month of 2017. This means that, unlike the 2017 season, the 2019 season was characterized by frequent precipitation. The analysis of variance (Appendix A) revealed a highly significant effect of treatment (T), year (Y) and their interaction (T × Y) for FHB severity (%) and grain yield (GY) (*p* < 0.001). The thousand kernel weight (TKW) was only significantly affected by the treatment (T) (*p* < 0.001).

### 3.2. Effect of FHB Severity on Wheat Yield Components in Control Plants

FHB was spotted in both the non-inoculated and inoculated controls; however, the severity was lower in the non-inoculated control (Table 3). The inoculation of control plants resulted in the highest level of FHB severity in 2019, reaching up to 73%, and in the lowest one in 2017, which was up to 54%. In parallel, the disease induced a reduction in grain yield in all three seasons, and the most pronounced reduction was recorded in 2019, resulting in the lowest yield of wheat, counting for 1.46 Mg·ha^−1^, which represents a reduction rate of 21.8%. Furthermore, the FHB attack decreased the TKW in the three cropping seasons, with a higher reduction of TKW in 2019 (Table 4).

### 3.3. Effect of Seed Coating Treatments on FHB Severity and Yield Components

All the treatments showed a notable potential in controlling FHB under field conditions (Table 3), with a reduced rate of FHB severity ranging from 30 to 70%. In terms of biocontrol, the treatments *M. guilliermondii*, *T. harzianum* and Panoramix were more efficient in 2017, while the combination *M. guilliermondii-T. harzianum* was steadily efficient in all three cropping seasons. All treatments increased the wheat yield not only with regard to the inoculated control but also to the non-inoculated control. The increase rate ranged from 25 to 68% compared to the inoculated control. *T. harzianum* and the combination *M. guilliermondii-T. harzianum* was the most efficient treatment in increasing wheat yield in all three cropping seasons. Almost all treatments increased TKW as well; however, the increase was slightly significant, and the highest TKW, reaching around 52 g, was obtained from the plants treated with combination *M. guilliermondii-T. harzianum* and Panoramix in 2017.

### 3.4. Effect of Seed Coating Treatment and FHB Severity on Physiological Traits and Isotopic Composition of Leaves and Grains

The clustered Pearson correlation matrix of all data (Figure 2) had shown a positive correlation between traits in cluster 1: [SPAD, NDVI, GY, TKW, δ^15^Ngrain, δ^13^Cleaf, Cgrainδ^13^Cgrain and Ngrain], and a second positive correlation between traits in cluster 2: [δ^15^N_leaf_, canopy temperature, N_leaf_, Fusarium head blight severity, stomatal conductance and C_leaf_]. The impact of *F. culmorum* on plant status was observed in the negative correlation between FHB severity and the traits within cluster 1. The trait C_leaf_ was found to have insignificant correlations with almost all traits except for δ^13^C^leaf^, stomatal conductance and N_leaf_.

At the early milk stage, the inoculated control plants had significantly reduced green leaf area (NDVI) and leaf chlorophyll content (SPAD) when compared with the other treatments (Table 5 and Appendix A). The inoculation increased the canopy temperature and stomatal conductance. This was associated with a decline in leaf C and N content and in the isotope compositions of δ^13^C_leaf_ and δ^15^N_leaf_. The resultant impact of FHB at harvest included a reduction of yield and TKW, as previously mentioned, together with lower C_grain_, N_grain_, δ^13^C_grain_ and δ^15^N_grain_.

At the late milk stage, and compared to the inoculated control, all treatments resulted in lower δ^15^N_leaf_ (^15^N/^14^N_leaf_) content. Differently, the treatments Panoramix, *T. harzianum* and the combination *M. guilliermondii-T. harzianum* increased leaf carbon content, and δ^13^C_leaf_(^13^C/^12^Cl_eaf_) decreased leaf nitrogen, while *M. guilliermondii* decreased leaf carbon content and δ^13^C_leaf_ (^13^C/^12^C_leaf_) and increased leaf nitrogen, compared to the inoculated control (Table 5).

At harvest, despite their different impact at the late milk stage, all the treatments resulted in higher C_grain_, δ^13^C_grain_ (^13^C/^12^C_grain_), N_grain_, and δ^15^N_grain_ (^15^N/^14^N_grain_), compared to the inoculated control (Table 5).

The combination *M. guilliermondii-T. harzianum* was the most stable treatment among the years and outperformed the other seed treatments by producing the highest grain yield and TKW, together with the highest NDVI, C_grain_, C_leaf_ and the lowest stomatal conductance, δ^13^C_leaf_, δ^15^N_leaf_, among treatments. The treatment that comes in second in terms of efficiency is *T. harzianum*, which resulted in the second-highest grain yield, highest NDVI, SPAD, N_leaf_ and δ^15^N_leaf_ values, the second-lowest stomatal conductance, and the lowest FHB severity and δ^15^N_grain_, among treatments

## 4. Discussion

### 4.1. Variability of FHB Severity and Its Impact on Physiological Traits, Yield Components and Stable Isotopic Composition

Northern Tunisia is an ecosystem favorable for fungal invasion in the field, which generally results in yield loss and quality reduction In Tunisia. The authors of [24] reported the occurrence of FHB disease on harvested durum wheat caused predominantly by *F. culmorum*. Previous investigations reported that Fusarium infection was considerably influenced by environmental conditions, especially temperature, rainfall and moisture during heading and flowering periods of cereals [25]. It was reported that the optimal conditions for the infection by *Fusarium* were the frequent rain, high humidity (92.94%) and temperature higher than 15 °C [21]. Likewise, in this study, the difference in the severity of *Fusarium* head blight among the years seemed relative to climatic conditions since the highest severity was under the highest and frequent precipitations and the lowest temperatures. These climatic conditions favor the survival and sporulation of the fungus on the cereal stubble debris.

Subsequently, the FHB attack exhibited a reduction in grain yield and TKW in the three cropping seasons. This supports the reported effect of *F. culmorum* on the reduction in yield components, 1000-kernel weight (TKW) and weight and number of kernels per head after inoculation [26]. These losses could be the result of premature bleaching of one or more infected spikelets in the cereal plant’s head [27] and/or sterility and poor grain filling caused by deterioration of transport of assimilates affecting grain composition [28,29], which all lead to shriveling or premature ripening of kernels [30].

The infection of plant tissue with fungal pathogens is closely linked to changes in metabolic pathways such as photosynthesis [9]. In this manner, our study demonstrates that the infection by FHB negatively affected the photosynthesis processes by decreasing the green leaf area (NDVI) and leaf chlorophyll content (SPAD) recorded at the grain filling. The findings of [31] corroborate our results, showing that *F. culmorum* decreased the photosynthetic efficiency of infected wheat ears.

Since no studies have reported the impact of FHB on the carbon and nitrogen metabolism of plants, a large part of this discussion was dedicated to gaining more insight from the results of carbon and nitrogen content and isotopic compositions in leaves and grains of non-treated plants under FHB attack. The impact of *F. culmoum* included a decline in leaf carbon and nitrogen content and in the isotope compositions δ^13^C_leaf_ and δ^15^N_leaf_, at grain filling stage, and in C_grain_, N_grain_, δ^13^C_grain_ and δ^15^N_grain_ at harvest (Figure 3).

According to the literature, the carbon content in grains is derived from photosynthetic fixation occurring during grain filling, from the diffusion of CO_2_ from the air into the leaves, through the stomata, and from the earlier-assimilated carbon remobilized from vegetative organs [32]. Through these enzymatic and physical processes, C3 plants discriminate against ^13^C in favor of ^12^C, leading to a lower ^13^C/^12^C ratio, thus a higher δ^13^C [32]. On this basis, the decline in δ^13^C_leaf_ and δ^13^C_grain_ indicates an increase in CO_2_ supply, which is supported by the increase in stomatal conductance. In fact, stomata constitute a passive gate for *F. culmorum* entry into the host tissue and have a crucial role since, through them, the fungus infects the spikelets internally by entering into the vascular bundles of the rachilla and rachis [33]. Considering the fact that pathogens have evolved virulence factors to counteract host stomatal defenses by either inhibiting stomatal closure or promoting stomatal opening [34], it might be assumed that *F. culmorum* has modulated the stomatal behavior of wheat for its own benefits.

On the other hand, the nitrogen content in grains is derived from direct nitrogen assimilation from roots during grain filling and from remobilization of earlier-assimilated nitrogen from vegetative organs to developing grains [35]. In general, during biochemical and physiological processes in plants, the heavier stable isotope ^15^N is discriminated against ^14^N, leading to a lower ^15^N/^14^N ratio, and thus, a higher δ^15^N [36]. The natural variation of the nitrogen isotope composition δ^15^N is linked to nitrogen sources used by the plant (NH^4+^ uptake will induce ^15^N enrichment compared to NO^3−^), to the activity of enzymes involved in the assimilation of N sources, to the nature of compounds resulting from nitrogen fractionation; proteins are generally ^15^N enriched compared to chlorophyll, lipids, amino sugars and alkaloids [37], and to volatilization, translocation or nitrogen recycling in the plant [35]. Thus, the decline in δ^15^N_leaf_ and δ^15^N_grain_ in this study indicates an increase of ^14^N fraction in both organs, which could reflect an enhanced NH^4+^ uptake and assimilation [38].

The contrasting decline in leaf carbon and nitrogen content occurring during grain filling and C_grain_ and N_grain_ content at harvest are assumed to be a consequence of the impact of pathogen infection on primary metabolism. In our case, there are two possible scenarios (i) a shift of N and C source allocations towards the requirements of defense-related pathways that would otherwise be used for grain filling [39], thus leading to lower N and C_grain_ storage; (ii) the pathogen tries to reroute the entire nutrients from the host to its advantage and thus could be considered a new sink for C sources as sugars and N sources as amino acids resulting in decreased N and C leaf and grain storage [39,40,41].

### 4.2. Effect of the Biostimulants on Physiological Traits

Biostimulants are considered as products modifying biochemical and physiological processes in plants, neutralizing the adverse impact of weather conditions and reducing the occurrence of diseases by stimulating plant growth, strengthening plant defense and improving nutrition efficiency leading to sustainable crop yield [24]. In our preceding experience under controlled conditions, the seed coating with *Meyerozyma guilliermondii* and *T. harzianum* S.INAT was found to achieve the seed priming state of seeds and induce plant resistance against *F. culmorum* [11,16]. Seed priming with a biostimulant is an effective pre-germination physiological method that induces structural and ultra-structural modifications and a change in plant hormone biosynthesis. The resulting signal could facilitate nutrient uptake and result in growth promotion, together with the induction of systemic resistance in challenged plants leading to an increase in crop yield in a sustainable manner [42]. In these terms, this study focused on evaluating the biostimulant potential of *T. harzianum*, *M. guilliermondii* and their combination, as well as Panoramix, when applied as seed coating agents, against FHB of durum wheat and the resulting grain yield and quality (carbon and nitrogen content) under field conditions.

All the treatments showed notable potential in controlling FHB severity under field conditions. Moreover, they all promoted the photosynthetic processes depicted by higher green leaf area (NDVI) and chlorophyll content (SPAD). The performance of *M. guilliermondii* and *T. harzianum* under field conditions aligned with our previous studies demonstrating their potential in inducing plant resistance against *F. culmorum* under controlled conditions [11,16]. In more detail, the seed coating with *T. harzianum* S.INAT was found to apply, under controlled conditions, to both the direct antagonist activity and indirect growth promotion and induced systemic resistance of wheat plants against foot crown rot disease [16,17]. As for *M. guilliermondii*, we report promotion of wheat growth and photosynthesis and control of *F. culmorum* attack [11].

Most importantly, all treatments improved wheat yield and grain quality by increasing the thousand kernel weight, C_grain_, N_grain_, δ^13^C_grain_ and δ^15^N_grain_. The increased thousand kernel weight reflects the resistance to kernel infection by FHB, as demonstrated by [43]. These results point to an enhanced N and C uptake and assimilation during vegetative growth and remobilization during grain filling [20]. These results could be considered as the outcome of both the growth promotion impact of these treatments and their ability to mitigate the FHB adverse effects on N and C metabolism.

Despite their common positive impact on wheat end-product, the three treatments, *T. harzianum*, Panoramix and the combination *M. guilliermondii-T. harzianum* showed a different impact from that of *M. guilliermondii* at the early and late milk stages. This calls attention to a difference in their mode of action. The three treatments, *T. harzianum*, Panoramix and the combination *M. guilliermondii*-*T. harzianum* behaved similarly but differed in terms of levels. Unlike M. guilliermondii, the three treatments Panoramix, *T. harzianum* and the combination *M. guilliermondii-T. harzianum* decreased stomatal conductance compared to the inoculated control, which reflects a reduction in stomatal opening. In fact, stomatal closure is one of the first lines of defense against pathogen invasion and is involved in the resistance to FHB in wheat by intense cross-talk pathways regulating the guard cell movements in order to restrict pathogen entry into the leaves [34]. Therefore, the three treatments Panoramix, *T. harzianum*, and the combination *M. guilliermondii-T. harzianum* are most likely able to counter the behavior of *F. culmorum* related to stomata openings and to enhance the plant's innate immune response. Moreover, the impact of these three treatments on lowering CO_2_ supply (higher δ^13^C_leaf_) could only be the limiting result of stomatal closure. There are two hypotheses that could explain their impact on increasing leaf carbon content compared to control, despite the restrained CO_2_ supply. The first is their ability to control FHB, thus minimizing the fungal mass and subsequently reducing the redirection of nutrients from the host to the pathogen leading to higher C_leaf_ content. The second is their ability to induce systemic resistance characterized by the accumulation of carbon-rich metabolites as carbohydrates, fatty acids and phenylpropanoids, which play a crucial role in wheat resistance [23], thus resulting in higher C_leaf_ content. The second hypothesis is backed up by our previous studies reporting that phenolic compounds and chitinase (involved in carbohydrate metabolism were highly induced in plants treated by *Trichoderma* and Panoramix in response to the Fusarium infection under controlled conditions [16,17]. Another common impact between the three treatments, Panoramix, *T. harzianum* and the combination *M. guilliermondii-T. harzianum*, is the increase of ^14^N uptake and assimilation (lower δ^15^N_leaf_) associated with a lower leaf nitrogen content compared to control. It is suggested that the three treatments redirected plant metabolism towards the allocation of resources for the onset of defense reactions and biosynthesis of protective compounds as flavonols (carbon-based secondary compounds) rather than towards the process of protein synthesis and photosynthesis (nitrogen-containing molecules) [44], which corroborates the observed increase of leaf carbon content mentioned earlier.

Contrastingly, compared to control, the impact of *M. guilliermondii* on increasing CO_2_ supply (lower δ^13^C^leaf^) could be attributed in part to the absence of defense-related stomatal closure. In addition, its impact on decreasing leaf carbon content compared to control, despite the higher CO_2_ supply, suggest that *M. guilliermondii* redirected plant metabolism towards the allocation of resources for the process of protein synthesis and photosynthesis (nitrogen-containing molecules) rather than towards the onset of defense reactions and biosynthesis of protective compounds as flavonols (carbon-based secondary compounds) [44]. The impact of *M. guilliermondii* on increasing ^14^N uptake and assimilation (lower δ^15^N_leaf_) and accumulation of leaf nitrogen could not agree more with the last assumption. Indeed, our previous report dealing with *M. guilliermondii* stated that, under infection, it promoted photosynthesis and transpiration traits, together with higher nitrogen/carbon ratio and lower flavonols, anthocyanins and ABA in leaves [11].

The combination *M. guilliermondii*-*T. harzianum* tended to be the most stable treatment among the years and outperformed the other seed treatments by producing the highest grain yield and TKW, together with the highest NDVI, C_grain_ and C_leaf_, and the lowest stomatal conductance, δ^13^C_leaf_, δ^15^N_leaf_, among treatments (Figure 3). This suggests that the outperformance of the combination *M. guilliermondii-T. harzianum* could be credited in part to the enhanced carbon metabolism and could be linked to the difference between *M. guilliermondii* and *T. harzianum* in the triggered pathways involved in growth promotion, systemic resistance, photosynthesis and nutrient metabolism. These differences are thought to be synergetic or complementary and probably cover the side effects from each treatment when applied solo and set off a balanced state within the plant. More precisely, it is suggested that the limiting result of *T. harzianum* on stomatal closure and lowering CO_2_ supply could be compensated by the absence of stomatal closure in *M. guilliermondii*-treated plants. Moreover, at the leaf level during grain filling, the accumulation of carbon-rich metabolites and the lack of nitrogen accumulation triggered in *T. harzianum*-treated plants could complement the accumulation of nitrogen-containing molecules and the lack of carbon accumulation triggered in *M. guilliermondii*-treated plants.

The treatment that comes in second in terms of efficiency is *T. harzianum*, which resulted in the second-highest grain yield, the highest NDVI, SPAD, N_leaf_ and δ^15^N_leaf_ values, the second-lowest stomatal conductance, and the lowest FHB severity and δ_15_N_grain_, among treatments. This suggests that a major part of the efficiency of *T. harzianum* is owing to the enhanced nitrogen metabolism.

## 5. Conclusions

This study revealed that the seed coating with either *T. harzianum* or *M. guilliermondii* showed effectiveness in counteracting the deleterious effects of the FH Band, promoting the wheat grain yield up to a range of 25 to 68%. The impact of treatments also included the promotion of photosynthesis (based on the data of NDVI and chlorophyll content) and a higher uptake and mobilization of nitrogen and carbon towards the grains (based on the data of δ^13^Cgrain and δ^15^Ngrain). We emphasize that the seed coating with combination *T. harzianum-M. guilliermondii* treatment showed a high potency of biological control, grain yield improvement and thousand kernel weight, suggesting that the two microorganisms have synergetic or complementary impacts on wheat. This suggestion underlines the need to decipher the mode of action applied by the combination through advanced molecular approaches. The beneficial interaction between the combination and variety “Karim” of durum wheat gives insight into the worth of further studying the effects of the combination on other varieties of wheat in order to seek better interactions.

## Figures and Tables

**Figure 1 microorganisms-09-02410-f001:**
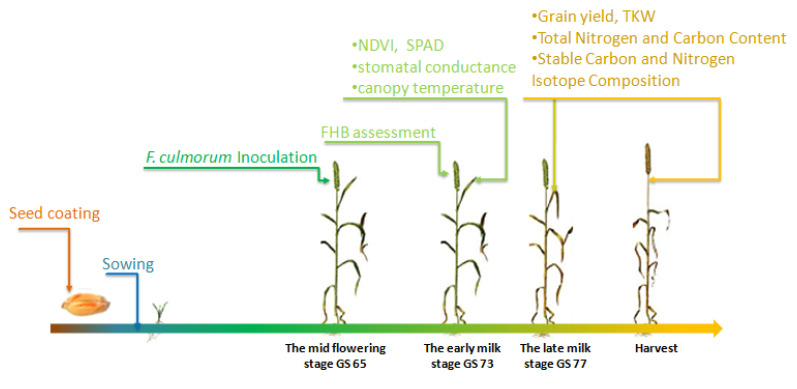
Illustration of the timeline of the field experiment; six manipulative steps are shown along the experiment timeline: (1) seed coating with 3 treatments (*M. guilliermondii*; the combination *M. guilliermondii-T. harzianum*; “Panoramix”) plus the non-inoculated and inoculated controls (each on 3 replicated pots) at one hour before sowing; (2) sowing; (3) aerial inoculation of wheat spikes with *F. culmorum* at the mid-flowering growth stage. (4) At 20 days post-inoculation (at the early milk stage), destructive sampling for disease assessment (100 heads per plot) and non-destructive measurements at the flag leaf level (NDVI, SPAD, stomatal conductance, canopy temperature); (5) collection of flag leaf samples for N/C content and isotopic compositions. (6) At harvest, quantification of grain yield, TKW, and grain sampling for N/C content and isotopic compositions. See methods for further details.

**Figure 2 microorganisms-09-02410-f002:**
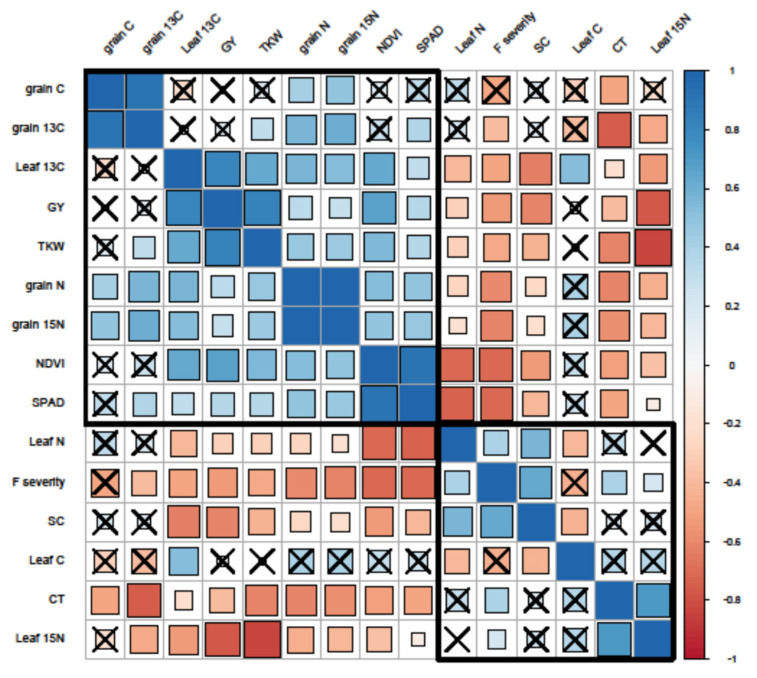
The Pearson’s correlation matrix among physiological traits, Fusarium head blight severity, yield components and grain stable isotope composition (2018 field trial). Traits; C: carbon, N: nitrogen, 13C: δ13C, 15N: δ15N, GY: grain yield, TKW: thousand kernel weight, CT: canopy temperature, F severity: Fusarium head blight severity, SC: stomatal conductance. The darker, bigger blue squares indicate a stronger positive correlation. The darker, bigger red squares indicate a stronger negative correlation. Crossed cells indicate statistically insignificant correlations. Black rectangles indicate the traits that have a similar pattern of correlation according to hierarchical clustering.

**Figure 3 microorganisms-09-02410-f003:**
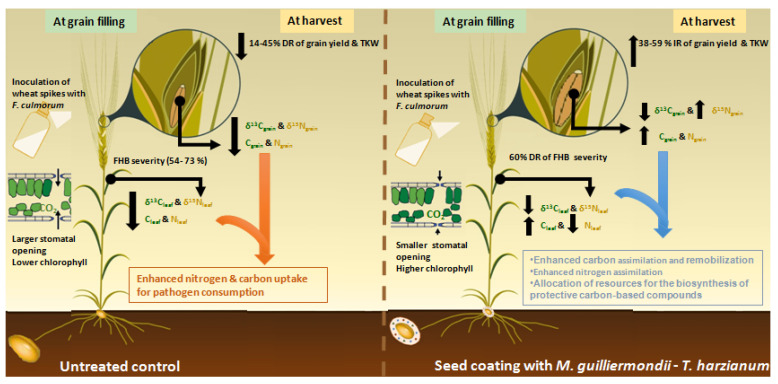
Schematic representation of the impact of FHB disease and the highest yielding treatment *M. guilliermondii-T. harzianum* on physiological traits, grain yield and N/C metabolism in durum wheat.

**Table 1 microorganisms-09-02410-t001:** The climatic conditions (temperature, precipitation and humidity) of the three years in the experimental station of Oued Beja.

Parameter	Mean Temperature (°C)			Precipitation (mm)		
Season	2016–2017	2017–2018	2018–2019	2016–2017	2017–2018	2018–2019
November	15.63	16.65	15.25	60	67.8	74
December	12.63	11.25	11	40.8	84.8	60.4
January	8.7	12.15	8	119.2	44.4	138.4
February	13.15	11.75	10.2	96.4	89.8	49.8
March	13.3	14.9	13.6	25.6	89.1	55
April	16.55	17.4	15.5	42.4	17.8	37
May	21.3	22.85	17.3	0	48.9	106.2
June	28.3	27.25	26.25	19.8	4.2	0
Sum/Average	16.195	16.775	14.6375	404.2	446.8	520.8

**Table 2 microorganisms-09-02410-t002:** The soil’s physicochemical characteristics of Oued Beja station.

pH			7.2		
Soil Type		Vertosol (Texture: Clay Loam)			
Composition of Soil					
Depth	Clay (%)	Loam (%)	Sand (%)	Mineral N (ppm)	Total N (%)
0–20	67.5	22.5	10	859	0.17
20–40	65	23.7	11.3	934.7	0.16

**Table 3 microorganisms-09-02410-t003:** Effect of treatments on FHB severity in durum wheat evaluated in a three-year study.

	Treatments					
Year	NIC	IC	CM	CP	CT	CC
2017	15.1 ± 0.7	54.3 ± 0.5	16.3 ± 1.5	17.0 ± 2.0	17.3 ± 0.5	22.5 ± 0.5
2018	23.3 ± 1.2	62.3 ± 0.5	38.6 ± 0.5	30.1 ± 1.0	22.1 ± 1.2	22.8 ± 0.7
2019	26.6 ± 0.2	73.1 ± 0.7	40.0 ± 1.0	40.1 ± 0.7	41.1 ± 1.0	24.6 ± 0.5
Comparison		IR (%)	RR compared to inoculated control (IC) (%)			
2017		72.1	69.98	68.7	68.1	58.5
2018		62.6	37.97	51.6	64.4	63.3
2019		63.6	45.32	45.1	43.7	66.29

NIC: non-inoculated-control; IC: inoculated control; CM: coated with *M. guilliermondii*; CT: coated with *T. harzianum*; CP: coated with Panoramix; CC: coated with combination *M. guilliermondii*-*T.*
*harzianum*; IR: increase rate; RR: reduction rate.

**Table 4 microorganisms-09-02410-t004:** Effect of treatments on yield components of durum wheat evaluated in a three-year study.

Traits		Treatments					
	Year	NIC	IC	CM	CP	CT	CC
Grain yield (Mg/ha)	2017	2.62 ± 0.62	2.17 ± 0.48	3.70 ± 0.16	3.40 ± 0.28	6.38 ± 0.01	6.82 ± 0.03
	2018	2.41 ± 0.17	2.01 ± 0.08	3.60 ± 0.23	2.70 ± 0.26	4.02 ± 0.17	6.37 ± 0.03
	2019	1.87 ± 0.17	1.46 ± 0.31	2.76 ± 0.25	2.86 ± 0.24	2.64 ± 0.12	2.77 ± 0.03
	Comparison		RR (%)	IR compared to inoculated control (IC) (%)			
	2017		17.3	41.4	36.3	66.0	68.2
	2018		16.6	44.1	25.5	49.9	68.3
	2019		21.8	47.1	48.8	44.7	47.3
Thousand Kernels weigh (TKW) (g)	2017	49.2 ± 0.2	48.1 ± 0.6	50.9 ± 0.2	51.9 ± 0.9	50.3 ± 0.2	52.6 ± 0.8
	2018	39.9 ± 0.6	38.8 ± 0.2	42.9 ± 1.4	42.1 ± 3.3	41.9 ± 0.8	45.9 ± 0.5
	2019	39.8 ± 0.7	37.7 ± 1.0	41.9 ± 0.9	36.2 ± 1.9	37.8 ± 0.9	39.0 ± 0.2
	Comparison		RR (%)	IR compared to inoculated control (IC) (%)			
	2017		2.2	5.4	7.2	4.2	8.5
	2018		2.9	9.6	7.9	7.4	15.5
	2019		5.4	10.2	−4.2	0.3	3.3

NIC: non-inoculated-control; IC: inoculated control; CM: coated with *M. guilliermondii*; CT: coated with *T. harzianum*; CP: coated with Panoramix; CC: coated with combination *M. guilliermondii*-*T.*
*harzianum*; IR: increase rate; RR: reduction rate.

**Table 5 microorganisms-09-02410-t005:** Effect of the biostimulants on the physiological traits and carbon and nitrogen isotope discrimination in the 2018 cropping season.

	**Yield Components**			**Physiological Traits**				
**GS**	**Harvest**			**the Early Milk Stage (GS 73)**				
**Traits**	**GY (Mg/ha)**	**TKW (g)**		**FHB (%)**	**NDVI**	**SPAD**	**CT (°C)**	**SC (mmol.m^−2^.s^−1^)**
NIC	2.41 ± 0.17	39.96 ± 0.68		23.33 ± 1.2	0.70 ± 0.003	45.19 ± 0.22	22.06 ± 0.4	276.8 ± 30.5
IC	2.01 ± 0.08	38.80 ± 0.26		62.33 ± 0.5	0.66 ± 0.040	42.39 ± 2.00	22.76 ± 0.5	249.9 ± 1.3
CM	3.60 ± 0.23	42.90 ± 1.42		38.66 ± 0.5	0.72 ± 0.017	45.81 ± 0.41	20.16 ± 0.5	276.8 ± 30.5
CP	2.70 ± 0.26	42.13 ± 3.31		30.16 ± 1.0	0.77 ± 0.018	48.94 ± 0.51	20.96 ± 0.3	249.5 ± 0.6
CT	4.02 ± 0.17	41.92 ± 0.89		22.16 ± 1.2	0.80 ± 0.003	50.49 ± 0.83	19.26 ± 0.05	220.0 ± 21.7
CC	6.37 ± 0.03	45.93 ± 0.58		22.83 ± 0.7	0.79 ± 0.006	46.80 ± 0.64	19.03 ± 0.2	212.3 ± 31.1
	The carbon and nitrogen content and isotope discrimination							
**GS**	**The late milk stage (GS 77)**				**Harvest**			
**Traits**	**C_leaf_(%, g DW)**	**δ^13^C_leaf_ (‰)**	**N_leaf_ (%, g DW)**	**δ^15^N_leaf_ (‰)**	**C_grain_ (%, g DW)**	**δ^13^C_grain_ (‰)**	**N_grain_ (%, g DW)**	**δ^15^N_grain_ (‰)**
NIC	40.58 ± 0.29	−29.14 ± 0.11	3.06 ± 0.057	4.26 ± 0.07	41.80 ± 0.11	−25.45 ± 0.01	2.06 ± 0.019	2.58 ± 0.07
IC	39.20 ± 0.38	−29.27 ± 0.04	2.73 ± 0.24	4.13 ± 0.06	39.19 ± 0.43	−25.75 ± 0.08	1.85 ± 0.074	1.38 ± 0.33
CM	37.03 ± 0.55	−29.42 ± 0.04	3.15 ± 0.079	2.51 ± 0.11	42.39 ± 0.28	−25.18 ± 0.05	2.00 ± 0.016	2.21 ± 0.2
CP	40.00 ± 0.004	−28.81 ± 0.09	1.70 ± 0.14	3.33 ± 0.39	41.26 ± 0.23	−25.31 ± 0.07	2.24 ± 0.028	3.28 ± 0.02
CT	39.79 ± 0.067	−28.95 ± 0.05	1.43 ± 0.006	4.01 ± 0.04	40.70 ± 0.16	−25.56 ± 0.06	1.94 ± 0.015	1.87 ± 0.11
CC	40.22 ± 0.11	−27.80 ± 0.23	2.19 ± 0.05	2.04 ± 0.12	40.39 ± 0.29	−25.49 ± 0.02	2.16 ± 0.008	2.89 ± 0.28

GS: growth stage; FHB: Fusarium head blight; CT: canopy temperature; SC: stomatal conductance; C_leaf_: leaf carbon content; N_leaf_: leaf nitrogen content; NIC: non-inoculated, control; IC: inoculated control; CM: coated with *M. guilliermondii*; CT: coated with *T. harzianum*; CP: coated with Panoramix; CC: coated with combination *M. guilliermondii*-*T. harzianum.* At the early milk stage, all the treatments resulted in higher NDVI and SPAD values and lower values of canopy temperature. Differently, the treatments Panoramix, *T. harzianum*, and the combination *M. guilliermondii-T. harzianum* decreased stomatal conductance compared to inoculated control, while no effect was observed for *M. guilliermondii* (Table 5).

## Data Availability

Data is contained within the article or Appendix A. The data presented in this study are available in [Appendix A].

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
