# Peer review of "Comparative Performances of Beneficial Microorganisms on the Induction of Durum Wheat Tolerance to Fusarium Head Blight"

_microorganisms, 2021, doi:10.3390/microorganisms9122410_

Round 1

Reviewer 1 Report

The MS microorganisms-1418577 with the title: Comparative performances of beneficial microorganisms on the induction of durum wheat tolerance to Fusarium head blight investigated the effect of seed coating with Trichoderma harzianum, Meyerozyma guilliermondii and their combination on (i) Fusarium head blight severity, durum wheat grain yield in three seasons, and (ii) on physiological parameters and the carbon and nitrogen content and isotope composition in leaves and grains of durum wheat. Here are my comments:

The ms has to be edited in terms of English by a native speaker of some other services.

Add “grain yield” in the Keywords

L36: the Latin name of durum wheat is WRONG! What authors wrote is the Latin name (Triticum aestivum L.) for the bread wheat. Please correct this mistake.

In the introduction section, the author should write the production of durum wheat  for the whole world as well as they do not have to pay attention for the local issue. Please do not focus on Tunisia.

The authors wrote the whole introduction in one paragraph. They should improve it and divide the introduction section into three paragraphs as follows: Durum wheat, Fusarium head blight disease, and the objectives.

Please cite this reference in the introduction and where it is possible: Hewedy, O.A.; Abdel Lateif, K.S.; Seleiman, M.F.; Shami, A.; Albarakaty, F.M.; M. El-Meihy, R. Phylogenetic Diversity of Trichoderma Strains and Their Antagonistic Potential against Soil-Borne Pathogens under Stress Conditions. Biology 20209, 189. https://doi.org/10.3390/biology9080189

Why the authors applied the treatments as coating method??

Authors should change ml to mL in the whole ms. Please check, particularly in material and methods

Authors should add latitude and attitude of the experiments’ place in L89

L91: pots??? How it can be?? Pots is for glasshouse or lab experiments, but what is the experimental units for the field? As the authors wrote their experiments were done in the field, so the word pots is WRONG!

The size of each plot was small! (1 × 3 m)

Figure 1: is it your own Figure? Or you modified it from somewhere? Cite it if it is not your own.

The authors should write their methods in details instead of the current version.

L171 leave space: Table3

Table 3: authors should divide table 3 into two tables or they can remove a part of it such as ANOVA because authors already add the means and SE.

How you used SPAD? And which time?

L404 figure>> Figure

Why the combination M. guilliermondii-T. harzianum was the most stable treatment among the years and outperformed the other seed treatments?

Why the authors did not use different doses or level for biological agents?

The authors should add the most important data in the text of the conclusion.

Reviewer 2 Report

The manuscript describes the results of research relating to the use of certain microorganisms as biocontrol agents against Fusarium pathogens in wheat cultivation. The presented research results are desirable and nowadays, biocontrol in the cultivation of cereal plants is becoming more and more important. There is little information on the use of Trichoderma in plant biocontrol in vivo. The results of these studies on the effectiveness of these agents seem to confirm the results of the few available scientific papers.
The manuscript is well prepared, easy to read, but requires editorial corrections before publication. 

Round 2

Reviewer 1 Report

L37-38: Please cite the following reference in the first sentence: https://doi.org/10.3390/plants10061040 

L69-71: . In addition, the use of chemical fungicides as seed coatings ..... their rapid natural degradation. Please cite this new and relevant ref. Ahmed, H.F.A.; Seleiman, M.F.; Al-Saif, A.M.; Alshiekheid, M.A.; Battaglia, M.L.; Taha, R.S. Biological Control of Celery Powdery Mildew Disease Caused by Erysiphe heraclei DC In Vitro and In Vivo Conditions. Plants 2021, 10, 2342. https://doi.org/10.3390/plants10112342

L116-121: Authors sometimes write μl, while in other places they write μL. As I commented before, authors should go through their ms and correct all these issues to be μL, mL etc.

Starting from Table 3- The authors wrote under the tables: the symbols indicate statistical significance (.,p< 0. 1; ***, p< 0.001). I do not see the values of P or the stars in the Tables, can authors explaine ? .,p< 0. 1; *** ?? what is written before P is wrong, please correct. Also, it should be P≤ 0.1 or P≤ 0.001 instead of p< 0. 1 or p< 0. 001.

Still there are some trpos error, author should correct all of them.
